

# Use of next generation sequencing to compare simple habitat and species level differences in the gut microbiota of an invasive and native freshwater fish species

Benjamin D. Gallo, John M. Farrell and Brian Leydet

Department of Environmental and Forest Biology, State University of New York College of Environmental Science and Forestry, Syracuse, NY, USA

## ABSTRACT

Research on the gut microbiome of host organisms has rapidly advanced with next generation sequencing (NGS) and high-performance computing capabilities. Nonetheless, gut microbiome research has focused on mammalian organisms in laboratory settings, and investigations pertaining to wild fish gut microbiota remain in their infancy. We applied a procedure (available at https://github.com/bngallo1994) for sampling of the fish gut for use in NGS to describe microbial community structure. Our approach allowed for high bacterial OTU diversity coverage (>99.7%, Good's Coverage) that led to detection of differences in gut microbiota of an invasive (Round Goby) and native (Yellow Bullhead) fish species and collected from the upper St. Lawrence River, an environment where the gut microbiota of fish had not previously been tested. Additionally, results revealed habitat level differences in gut microbiota using two distance metrics (Unifrac, Bray–Curtis) between nearshore littoral and offshore profundal collections of Round Goby. Species and habitat level differences in intestinal microbiota may be of importance in understanding individual and species variation and its importance in regulating fish health and physiology.

## INTRODUCTION

Bacterial communities inhabiting the alimentary canal of organisms, often referred to as the host's "gut microbiome", have become a focal area of research over the last decade (*Gallo, Farrell & Leydet, 2020*). Studies show that gut microbiota can greatly influence host growth and development (*Lozupone et al., 2012*), behavior (*Johnson & Foster, 2018*), and immune system function (*Colombo et al., 2015*). To date, the majority of gut microbiome research has focused on mammals as model organisms for understanding vertebrate microbial communities (*Sullam et al., 2012*). Mammals though, comprise a relatively small proportion of the total vertebrate diversity, whereas fish represent ~50% (*Sullam et al., 2012*). Nonetheless, little is known surrounding the ecology of host-borne

Corresponding author
Brian Leydet, bfleydet@esf.edu

microbes in fish, particularly what factors drive patterns of bacterial colonization and community assemblage (*Tarnecki et al., 2017*). Because an organism's gut microbiome can influence many aspects of host physiology, describing the relative abundance of various microbes is an important first step in delineating organisms and/or communities that either benefit or harm the host.

Several factors are known to modulate gut microbiota composition in fishes, including host species/genetics (*Li et al., 2012*, *2014*), feeding habits (*Michl et al., 2017*), trophic levels (*Liu et al., 2016*), disease prevalence in the host population (*Hennersdorf et al., 2016*), and environmental variables including habitat and husbandry practices (*Dehler, Secombes & Martin, 2017*; *Wu et al., 2012*). Recent research on euryhaline fish has indicated that habitat salinity also plays a significant role influencing the dominant gut microbiota (*Schmidt et al., 2015*). Additionally, laboratory studies investigating the gut microbiota of Zebrafish *Danio rerio* demonstrate taxonomic similarities despite being raised in different aquaculture facilities (*Roeselers et al., 2011*). However, Zebrafish gut microbiota was also shown to differ temporally during ontogeny, highlighting the dynamic nature of gut-borne microbial communities (*Stephens et al., 2016*). Additional research is warranted to understand how gut microbial communities develop in nature in order to elucidate the beneficial and deleterious interactions between gut microbes and the fish host.

The objective of this study was to adapt a mammalian-based gut microbiome sampling and sequencing protocol to explore the gut microbiota from two fish species from the upper St. Lawrence River. We collected fish mucosal digesta to test for differences in the autochthonous gut microbiome of Round Goby (*Neogobius melanostomus*) and Yellow Bullhead (*Ameiurus natalis*). Yellow Bullhead, a native species to the upper St. Lawrence River, consumes small fish and crustaceans (*Stegemann, 1989*) whereas the invasive Round Goby diet is often dominated by invasive Zebra Mussels *Dreissena polymorpha* (*Ray & Corkum, 1997*). Furthermore, Round Goby generally prefer hard substrate in both shallow and deepwater habitats (*Charlebois et al., 2001*) while Yellow Bullhead prefer soft substrates in vegetated areas of shallow lakes, reservoirs and streams (*Stegemann, 1989*). These diet and habitat differences, in addition to known interspecies variation in fish gut microbiota (*Li et al., 2012*) provide a scenario for testing expected differences with the NGS workflow.

## MATERIALS AND METHODS

### Data collection and field processing

Collection of fish took place on the upper St. Lawrence River and its tributaries in Clayton, NY, USA (Fig. 1). All specimens were collected through volunteer angling, baited minnow traps, or fine mesh hoopnets and under permit from the New York State Department of Environmental Conservation (license #354). Capture date, total length (mm) and total weight (g) were recorded for each captured fish (Table S1). Euthanasia of animals followed approved protocols outlined by the American Veterinary Association and the American Fisheries Society through an overdose of Tricaine methansulfonate (400+ mg/L) or blunt cranial concussion (State University of New York College of Environmental

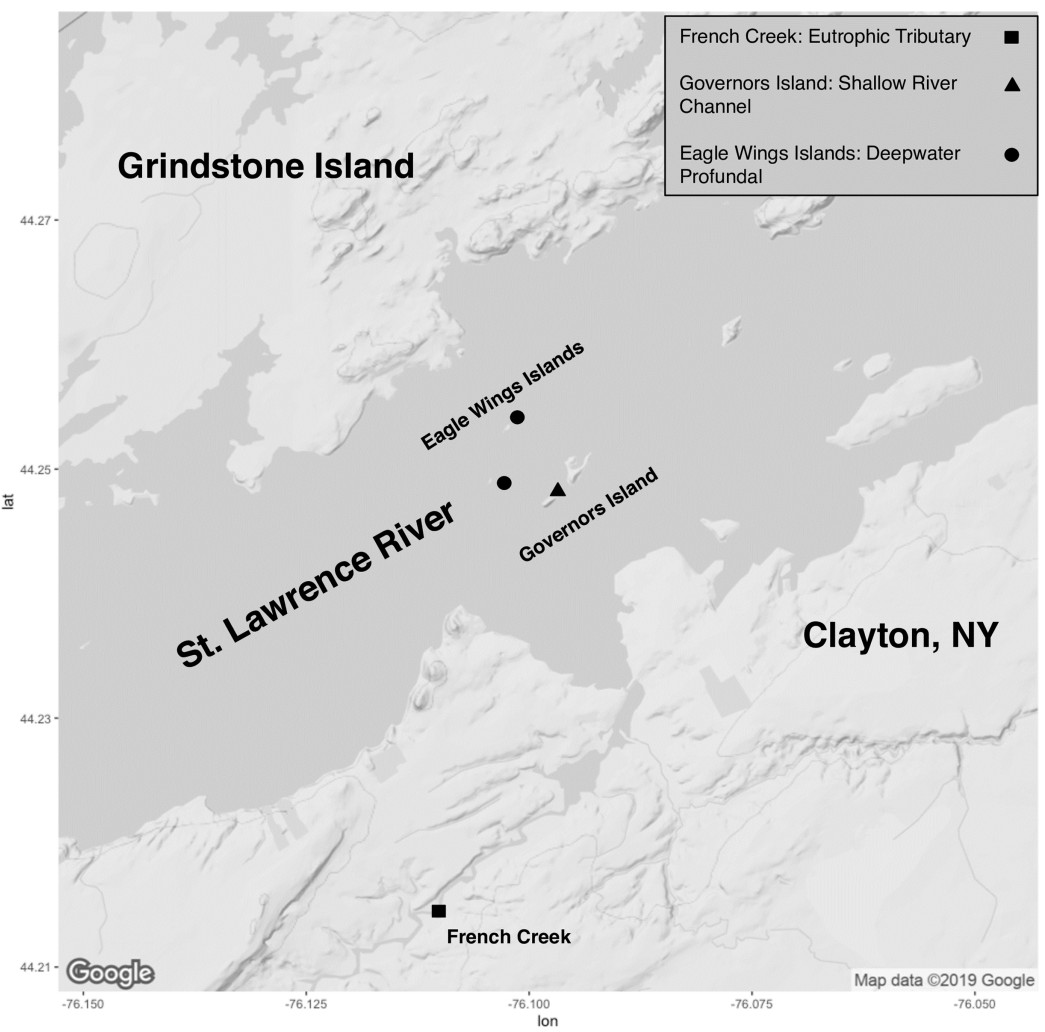

**Figure 1 Sampling locations along the St. Lawrence River (Clayton, NY).** Round Goby were sampled at Governors Island (blue: RG30-RG45) and the Eagle Wings Islands (white: RG25-RG29). Yellow Bullhead were sampled at French Creek (red: YBH39-YBH56). Circles denote approximate sample location (Image generated using ggmap: Kahle & Wickham, 2013). Map credit: ©2019 Google.

Science and Forestry's Institutional Animal Care and Use Committee protocol #180202). All fish were euthanized within approximately 3 h of capture and minnow traps/hoopnets were allowed to soak for a maximum of 18–24 h. There was no mixing of species or within species by sampling location prior to euthanasia to ensure no sharing of water and potential transfer of microbiota. Fish gut microbiota comparisons involved eight Round Goby captured at Governors Island (shallow littoral: <2 m water depth) and seven Round Goby at the Eagle Wings Islands (deepwater profundal: >15 m water depth). Seven Yellow Bullhead were also sampled in French Creek (coastal wetland tributary: <1 m water depth). Due to sampling habitat limitations for Yellow Bullhead, sampling took place within only one site, and thus Yellow Bullhead were only utilized for inter-species analyses. Round Goby samples from the two separate habitats (Governors Island and Eagle

Wings Islands) were combined for inter-species gut microbiota analysis but analyzed separately when testing for inter-habitat gut microbiota associations.

After euthanasia, the integument of each fish was surface sterilized by rinsing in a bath of 95% ethanol prior to dissection. All sample collection methods took place at room temperature (~20 °C). Fish were dissected with a posterior incision near the pectoral fin origin to the urogenital opening. Approximately 25 mg of hindgut tissue was aseptically removed from each fish using flame-sterilized dissecting scissors and/or scalpels. Digesta was manually cleared from the intestinal tract. Samples were then gently washed with a stream of sterile 0.05M phosphate-buffered saline (PBS) removing residual digesta while leaving autochthonous bacteria in the hindgut. All samples were stored in 1.75 mL of nucleic acid preservation buffer (NAP; 0.019 M ethylenediaminetetraacetic acid disodium salt dihydrate, 0.018 M sodium citrate trisodium salt dehydrate, 3.8 M ammonium sulfate, pH 5.2) allowing for ambient room temperature storage in sterile 2.0 mL microcentrifuge tubes (*Camacho-Sanchez et al., 2013*). Samples were stored for 4–8 weeks prior to DNA extraction. Previous examination of NAP buffer showed similar, if not superior, bacterial DNA preservation compared to commonly employed commercial buffers in comparative microbiome analyses (*Menke et al., 2017*).

## Laboratory processing

DNA was extracted using the E.Z.N.A® Tissue DNA Kit (Omega Bio-Tek, Norcross, GA, USA) following the manufacturers protocol, except that an overnight (15–17 h) tissue lysis step was employed to assist in complete intestinal tissue digestion. Extracted DNA was stored at −20 °C until PCR was performed. To selectively amplify bacterial DNA extracted from the hindgut samples, PCR was conducted using 16S V6-V8 rRNA primers (B969F and BA1406R (*Comeau et al., 2011*); Integrated DNA Technologies, Coralville, IA, USA) fused with unique barcodes and Illumina® adapter sequences. Employing this method (primers fused with NGS barcodes and adaptor sequences) resulted in significant cost savings and eliminated further sample processing and amplification steps (see *Comeau, Douglas & Langille, 2017*). The PCR master mix was created following *Comeau, Douglas & Langille (2017)* with the slight modification of using Q5® High-Fidelity Taq polymerase (New England Biolabs, Ipswich, MA, USA). The PCR cycling protocol was as follows: initial denaturation of 95 °C for 30 s, followed by 35 cycles of 95 °C for 30 s, 55 °C for 30 s, 72 °C for 30 s, and a final extension step of 72 °C for 5 min. PCR amplicons (~600 bp) were verified by gel electrophoresis on a 2% agarose gel. Amplicon cleanup and NGS preparation took place using the Agencourt AMPure XP PCR purification kit (Agencourt Biosciences, Beverly, MA, USA) and following manufacturer's protocol. NGS libraries were quantified via the Quant-iT dsDNA HS Assay (Invitrogen, Carlsbad, CA, USA) following the assay's standard protocol. Fluorescence was read on a Biotek® Synergy 2 plate reader (Agilent Technologies, Santa Clara, CA, USA) and samples were subsequently converted from $ng^*\mu l^{-1}$ to nM. The NGS library was diluted to a final concentration of 4 nM and the normalized libraries were pooled. For loading on the sequencer, 5 μl of the library pool was added to 5 μl freshly prepared 0.2 M sodium hydroxide, mixed well and incubated at ambient temperature for 5 min. This was followed

by an addition of 990 µl of pre-chilled Illumina® HT1 buffer, creating a final 20 pM library concentration. The prepared library was sequenced on an Illumina® MiSeq™ (San Diego, CA, USA) using 2 × 300 v3 chemistry and a 10% PhiX spike at the SUNY Molecular Analysis Core (SUNYMAC) at SUNY Upstate Medical University (http://www.upstate. edu/sunymac/).

## Dataset organization for analyses

For all described analyses, Round Goby and Yellow Bullhead samples were organized into two distinct datasets: (1) gut microbiota vs. fish species (15 Round Goby vs. 7 Yellow Bullhead) and (2) gut microbiota vs. fish habitat (7 Eagle Wings Islands Round Goby vs. 8 Governors Island Round Goby). All quality filtering, OTU clustering and multivariate comparison procedures were identical between group analyses.

## Sequenced data processing and analysis

Raw reads have been deposited with links to BioProject accession number PRJNA528762 in the NCBI BioProject database (https://www.ncbi.nlm.nih.gov/bioproject/). Sequenced libraries were demultiplexed in MiSeq™ Reporter v2.6 and FASTQ files were processed using USEARCH v.11.0.667 (Edgar, 2013). FASTQ sequences were stitched and filtered to the approximate size of the V6-V8 region of the bacterial 16S rRNA gene (400–600 bp length; Comeau, Douglas & Langille, 2017). USEARCH was used to trim primer regions and remove chimeric and low-quality sequences. Next, reads were merged based on similarity (400–600 bp in length, had ≤5 nucleotide differences, and were ≥90% similar). Sequences were filtered with the maximum expected errors per sequence ≤0.5. These cutoffs followed default or more stringent parameters as outlined in the USEARCH guide (https://www.drive5.com/usearch/). Filtered reads were subsequently preclustered by size (99% similarity, maximum differences ≤4) and then clustered into Operational Taxonomic Units (OTUs) based on 97% similarity using the UPARSE algorithm (Edgar, 2013). USEARCH filtering and subsequent OTU clustering was conducted with singleton data (OTUs with single DNA sequence occurrence), as procrustes analysis revealed non-significant differences between singleton and non-singleton multivariate ordinations (package "vegan", function "protest()", Procrustes analysis: SS = $2.47 \times 10^{-4}$ (Species Comparison) and $1.66 \times 10^{-4}$ (Habitat Comparison), $P \leq 0.001$ (Species Comparison) and $P \leq 0.001$ (Habitat Comparison); Figs. S2 and S3). All USEARCH scripts utilized in these analyses can be retrieved from our GitHub page (https://github.com/bngallo1994)

The USEARCH generated OTU tables were modified into a shared compatible file and uploaded into Mothur v.1.39.5 (Schloss et al., 2009). Rarefaction files were subsequently created using the Mothur MiSeq™ Standard Operating Procedure (https://www.mothur. org/wiki/MiSeq_SOP). OTU tables were modified to make rarefaction files to estimate species richness. Because of error rates, singleton calls using NGS platforms can be interpreted as potential sequencing artifacts (Brown et al., 2015). Nonetheless, subsequent analysis from the most dominant OTU's in each dataset revealed negligible differences in community structure between singleton-included and excluded microbiota matrices

(data not shown). Rarefaction curves utilizing singleton OTUs thus served as a maximum estimate of total species richness are presented in this manuscript.

The SILVA 138 high quality ribosomal RNA database was used to determine the identity of the most abundant OTUs in each fish species (*Quast et al., 2013*). We queried all OTU16S rRNA sequences using SILVA's Alignment, Classification and Tree Service (ACT) web module (https://www.arb-silva.de/aligner/) (*Pruesse, Peplies & Glöckner, 2012*). OTU sequences were classified with a minimum query sequence identity of 95% and 10 neighbors per query sequence.

The SILVA generated taxa information was subsequently combined with the USEARCH OTU table and sample metadata into a single phyloseq class object using the bioconductor package "Phyloseq" (*McMurdie & Holmes, 2013*). Two phyloseq objects were created—one for the Round Goby vs. Yellow Bullhead Species comparison, and one for the Governors Island vs. Eagle Wings comparison for Round Goby. Additionally, we analyzed the top 10 OTUs from each dataset at normalized sequencing depth (species comparison: 41,250 sequences/sample; habitat comparison: 40,972 sequences/sample) to determine their average relative abundance in Round Goby and Yellow Bullhead. Using the same strategy, we determined the identity and relative abundance of the top 10 OTUs within each Round Goby habitat group (Eagle Wings Islands vs. Governors Island).

To estimate the microbial community coverage, OTU tables and rarefaction data files were analyzed using the entropart (v 1.5-3; *Marcon, 2018*), ggplot2 (v 3.0.0; *Wickham, 2016*) and vegan (v 2.5.2; *Oksanen et al., 2018*) packages of the R statistical software (*R Core Team, 2017*). Rarefaction analyses were paired with the calculation of Good's Coverage using the Chao method to provide a description of rarefaction in terms of the total sampled OTU diversity (*Good, 1953*; *Chao, 1984*; *Larsen, Mohammed & Arias, 2014*). Datasets were rarefied to normalized sequencing depth for comparisons of microbial relative abundances between samples. The instantaneous slope of each rarefaction curve was determined at this normalized depth (*Hurlbert, 1971*). To estimate OTU loss from normalization, we also compared the normalized species richness to the observed richness collected at each sample's maximum sequencing depth (*Chao & Jost, 2012*). These analyses quantitatively assessed our ability to detect a representative coverage of microbial DNA from wild caught Round Goby and Yellow Bullhead.

A number of α-diversity metrics were also calculated for all Round Goby and Yellow Bullhead samples. The Observed, Chao1, Shannon and Simpson indices were calculated to estimate α-diversity between the two fish species and within Round Goby by habitat. Significance between groups was tested from each index with Wilcoxon rank-sum tests (Mann–Whitney) using the *Benjamini & Hochberg (1995)* P-value correction method.

Non-metric multidimensional scaling (NMDS) was employed to visualize dissimilarity in datasets between samples. NMDS (R package "vegan", function "metaMDS") ordination was computed for both normalized habitat and species datasets. NMDS scaling (*Kruskal, 1964*) was used to visualize the Bray–Curtis dissimilarity (*Bray & Curtis, 1957*) between samples using default permutations ($n = 20$) with the "metaMDS" function in the vegan package. Permutational multivariate analysis of variance using distance matrices (PERMANOVA, package "vegan", function "adonis"; 999 permutations) was

employed to assess significant differences (α = 0.05) between the two datasets: (species comparison = Round Goby and Yellow Bullhead; habitat comparison = Round Goby shallow and deeper profundal habitat). Due to the sample size differing by one individual, we assumed a balanced design and followed the recommendations of *Anderson & Walsh (2013)*; whereas PERMANOVA remains robust in the presence of heterogeneity of group dispersion.

In addition to NMDS, β-diversity was compared between fish species and habitat using pairwise weighted UniFrac distances (*Lozupone & Knight, 2005*). Unifrac distances were calculated between all combinations of Round Goby and Yellow Bullhead (for the species comparison) and Governors Island and Eagle Wings Round Goby (for the habitat comparison). Similar to the NMDS ordinations, weighted Unifrac distances were visualized using Principal Coordinate Analysis (PCoA) while retaining the first two axes. Welch's two sample *t*-tests were conducted to determine if weighted Unifrac distances between sample groups was significant. These distances were subsequently plotted (±SE) to visualize differences in gut microbiota by fish species and habitat.

Predictions of microbial community functions were assessed using the "Tax4Fun" (*Aßhauer et al., 2015*) and "themetagenomics" (*Woloszynek et al., 2019*) R packages. 16S rRNA marker gene functions were linked to SILVA database Kyoto Encyclopedia of Genes and Genomes (KEGG) orthologs (KOs) using the MoP-Pro approach to determine predictive function relative gene abundance for each gut microbiota sample (*Aßhauer & Meinicke, 2013*). The top 20 KO's were then screened (*Yang et al., 2019*) and plotted on bar graphs to related relative gene abundance vs. fish species or capture location (for Round Goby). Orthologs that did not link to a specified level one KEGG pathway were excluded from the analyses, and the next most abundant was substituted. Functions were combined based on level one as defined by the KEGG pathways output. Significance testing comparing the gut microbiota samples was performed using Welch's two sample *t*-tests.

## Supplementary data processing and analysis information

All procedures including tissue sampling, DNA extraction, PCR, amplicon cleanup, NGS library preparation and downstream FASTQ processing as well as all code are detailed on our GitHub page (https://github.com/bngallo1994).

# RESULTS

## NGS sequencing

Species and habitat comparison datasets yielded 3.01 and 1.90 million paired-end sequenced reads, respectively, each passing our pre-defined USEARCH quality filter. Final reads clustered into 1,266 OTUs and 574 OTUs in species and habitat comparison OTU tables, respectively. Rarefaction analyses (Figs. 2A and 3A) indicated plateauing of OTU detection at approximately 20,000–30,000 reads/sample, but novel OTU's were still detected at upwards of 100,000 reads/sample, especially in Yellow Bullhead. The mean number of reads/sample ±SE was 136,959 ± 776 with a mean of 126,739 ± 967 for Round Goby and 158,858 ± 1,283 in Yellow Bullhead. Analysis of rarefaction data using

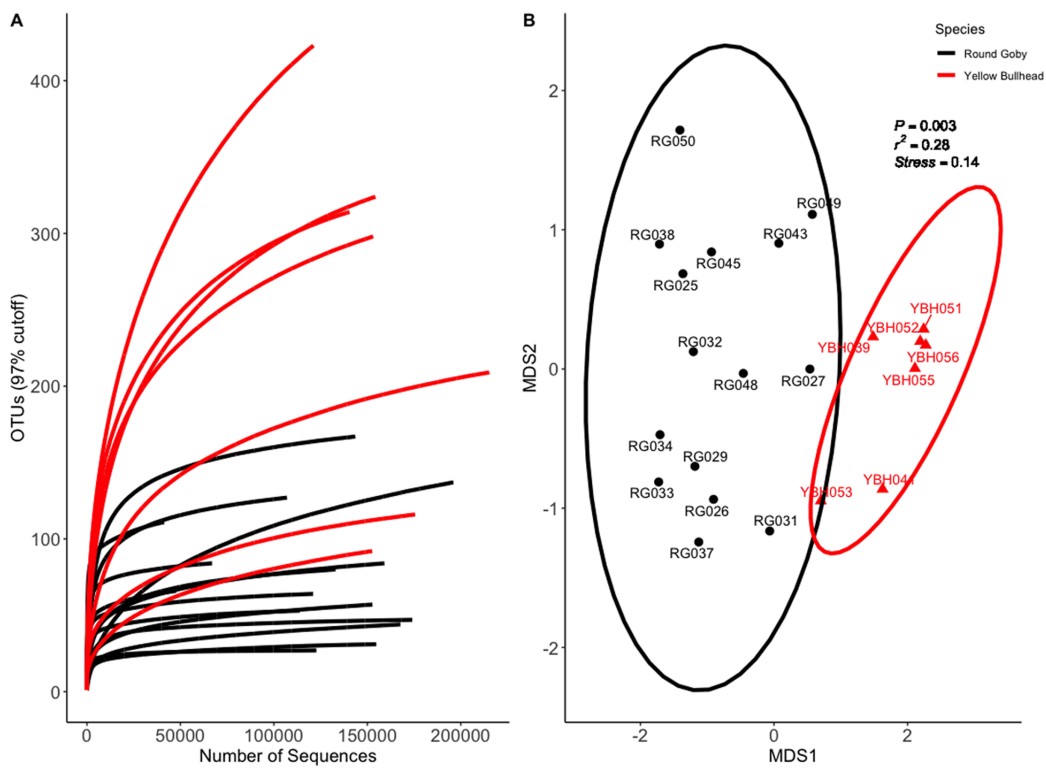

**Figure 2 Rarefaction Curve (A) and NMDS ordination (B) for species comparison between Round Goby (RG) and Yellow Bullhead (YBH).** Ellipses on the NMDS plot denote 95% confidence intervals. The NMDS ordination revealed a significant difference ($P = 0.003$) between the two species.

Good's Coverage at the normalized sequencing depth indicated samples represented >99.7% coverage of OTU diversity in both datasets (Table 1).

Slope analysis of normalized rarefaction curves indicated high rates of microbe OTU detection at low sequencing depth and low novel OTU detection at higher sequencing depth. Discovery rates of OTUs/1000 sequences show the slope of the normalized rarefaction curves were between 0.091–2.66 OTUs/1000 sequences between both datasets. Further validation of our high OTU coverage was seen comparing the OTU species richness at the normalized sequencing depth to maximum sequencing depth for each sample. The calculated highest sequencing depth provided an average ± SE increase of 32 ± 14 OTUs and 11 ± 4 OTUs for the species and habitat datasets respectively (Tables S2 and S3).

## Comparing Round Goby and Yellow Bullhead Gut Microbiota (SPECIES) and Round Goby Microbiota at the Eagle Wings Islands and Governors Island (HABITAT)

Permutational analysis of variance (PERMANOVA) on the species comparison NMDS ordination (Fig. 2B) indicated a significant difference between Round Goby and Yellow Bullhead gut microbial communities (PERMANOVA: Pseudo $F = 7.88$, df = 1, $P = 0.003$, $r^2 = 0.28$, NMDS stress = 0.14). Two distinct groups are evident with 95% confidence

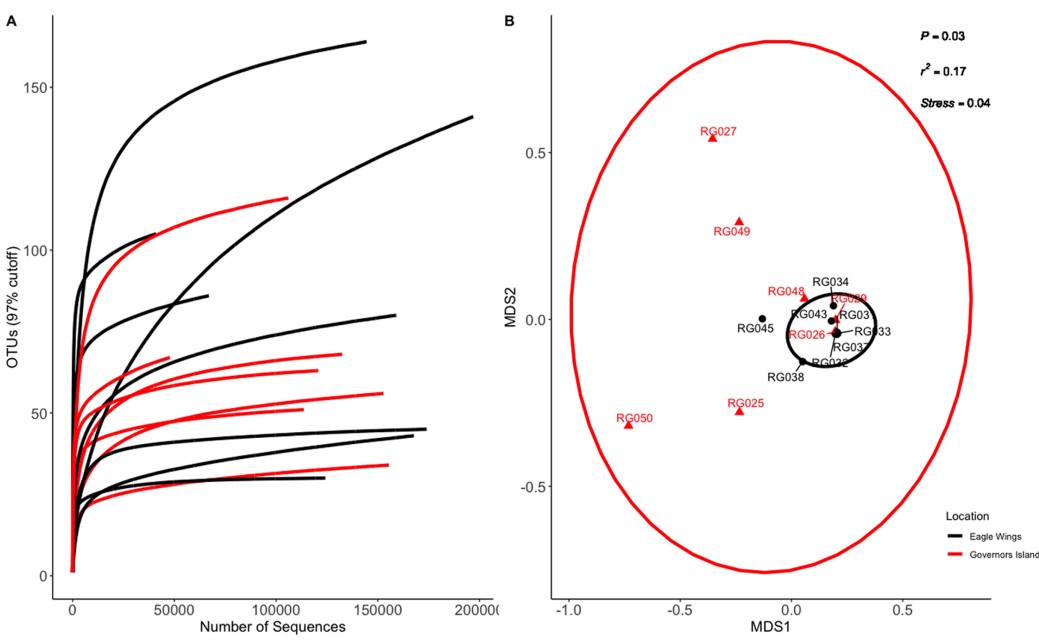

**Figure 3 Rarefaction Curve (A) and NMDS plot (B) for Round Goby (RG) habitat comparison between Governors Island and the Eagle Wings Islands.** Ellipses on the NMDS plot denote 95% confidence intervals. A significant difference ($P = 0.03$) was found between the Round Goby at each respective location.

ellipses displaying little overlap (Fig. 2B). Similar patterns were observed with the weighted Unifrac PCoA ordination (Axes 1 + 2 accounting for over 73% of total data variation (Fig. S3)).

OTU identification in both Round Goby and Yellow Bullhead also showed differences in the bacterial composition of dominant microbes in each of the fish species. Parsing the present microbiota by relative abundance, the top 10 OTUs (Fig. 4) using the SILVA database showed *Aeromonas* spp. (OTU1, 66.9 ± 8.7%), *Cetobacterium* spp. (OTU2, 10.1 ± 6.4%), and *Streptococcus* spp. (OTU5, 4.4 ± 3.3%), as the most abundant taxa in the gut microbiota of Round Goby. While *Cetobacterium* spp. (OTU2, 39.7± 14.7%), *Clostridium sensu stricto 1* (OTU7, 25.7 ± 8.7% and *Aeromonas* spp. (OTU1, 14.8 ± 9.4%) were the most abundant bacteria in Yellow Bullhead.

Similar analyses comparing Round Goby sampled between Governors Island and the Eagle Wings Islands (Fig. 3B) also revealed differences in their gut microbial composition (PERMANOVA: Pseudo $F = 2.63$, df = 1, $P = 0.03$, $r^2 = 0.17$, NMDS stress = 0.04). PCoA ordinations showed similarity to the NMDS plot (Fig. S4), and the plotted principal components accounted for over 89% of the data's variation. Additionally, differences were observed in the dominant gut microbiota from Round Goby at each location (Fig. 5). The Eagle Wing's Round Goby gut samples were dominated by *Aeromonas* spp. (OTU1, 88.9 ± 4.0% ), *Shewanella* spp. (OTU8, 4.2 ± 2.7%), and *Corynebacterium* spp. (OTU14, 1.1 ± 0.7%) while Governors Island Round Goby guts were largely colonized by *Aeromonas* spp. (OTU1, 47.3 ± 12.4%), *Cetobacterium* spp. (OTU2, 18.6 ± 11.6%) and

**Table 1  Good's coverage index calculations for all species (left) and habitat (right) datasets.**

| ID | Good's coverage |
|---|---|
| Species comparison | |
| RG025 | 1.000 |
| RG026 | 1.000 |
| RG027 | 1.000 |
| RG029 | 1.000 |
| RG031 | 0.999 |
| RG032 | 1.000 |
| RG033 | 1.000 |
| RG034 | 1.000 |
| RG037 | 1.000 |
| RG038 | 1.000 |
| RG043 | 0.999 |
| RG045 | 1.000 |
| RG048 | 1.000 |
| RG049 | 1.000 |
| RG050 | 1.000 |
| YBH039 | 0.999 |
| YBH041 | 0.999 |
| YBH051 | 0.997 |
| YBH052 | 0.998 |
| YBH053 | 0.999 |
| YBH055 | 0.998 |
| YBH056 | 0.999 |
| Habitat comparison | |
| RG025 | 1.000 |
| RG026 | 1.000 |
| RG027 | 1.000 |
| RG029 | 1.000 |
| RG031 | 0.999 |
| RG032 | 1.000 |
| RG033 | 1.000 |
| RG034 | 1.000 |
| RG037 | 1.000 |
| RG038 | 1.000 |
| RG043 | 0.999 |
| RG045 | 1.000 |
| RG048 | 1.000 |
| RG049 | 1.000 |
| RG050 | 1.000 |

**Note:**
Calculations were computed at the normalized singleton sequencing depth (41,250 sequences for all samples, 40,972 for RG only samples). All the samples revealed over 99.7% OTU diversity coverage. RG, Round Goby; YBH, Yellow Bullhead; ID, unique sample identification.

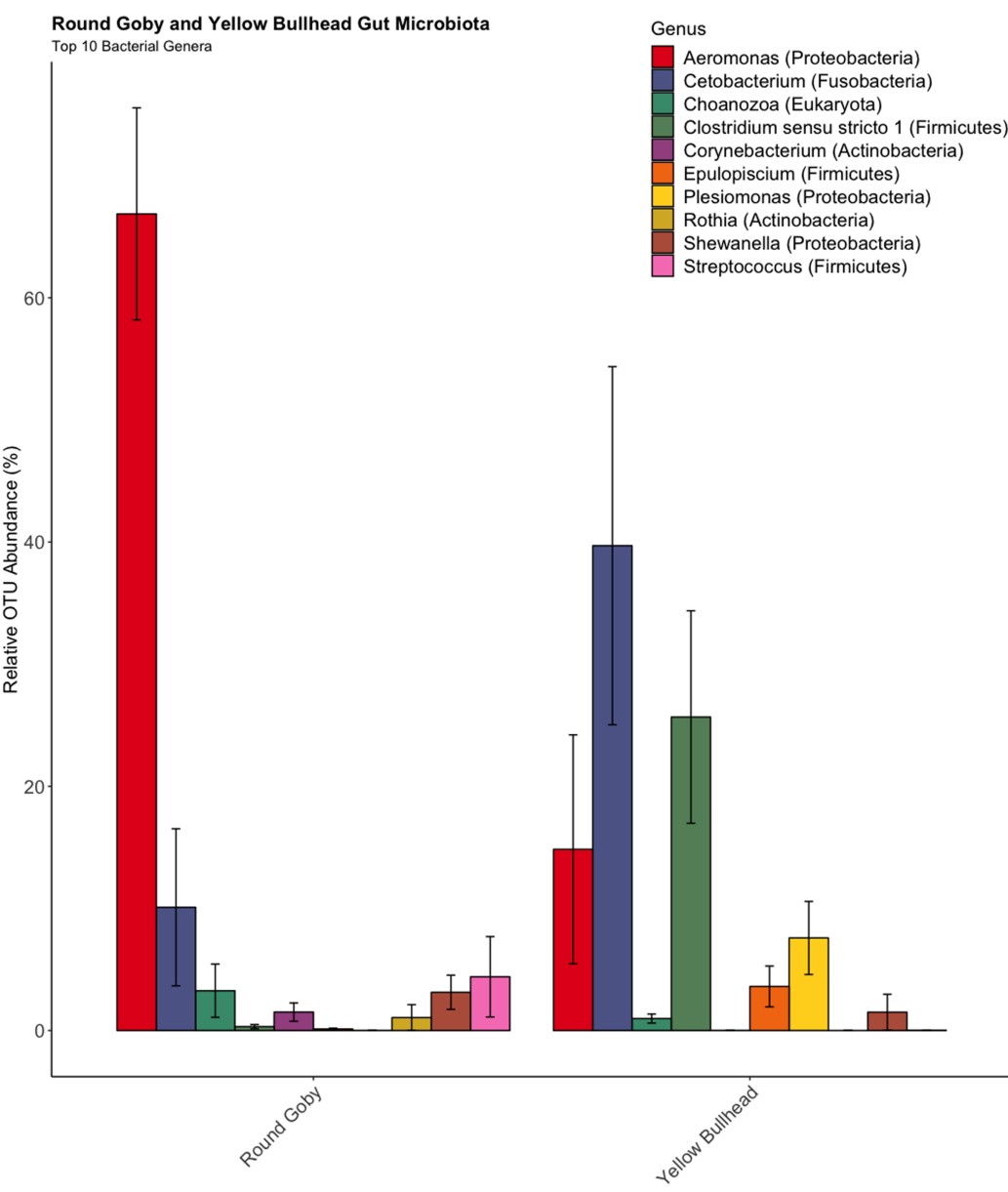

**Figure 4 Relative abundance of Top 10 OTU's for Round Goby and Yellow Bullhead from the species comparison.** *Aeromonas* spp., *Cetobacterium* spp., and *Streptococcus* spp. were the most abundant genera in Round Goby, while *Cetobacterium* spp., *Clostridium sensu stricto 1* and *Aeromonas* were most abundant in Yellow Bullhead.

*Streptococcus* spp. (OTU3, 8.1 ± 6.1%) bacteria. In both comparisons, the Top 10 dominant bacteria accounted for over 89% of the total microbiota observed.

Despite the observed differences in the dominant taxa between and within sampled fish gut microbiota, no significant differences existed for any α-diversity metrics, including the Observed, Chao1, Shannon and Simpson indices (*P*-values > 0.18) among species (Fig. 6). Similar trends were seen within Round Goby through the habitat comparison (*P*-values > 0.23; Fig. 7). UniFrac distances however, revealed higher variance in the gut

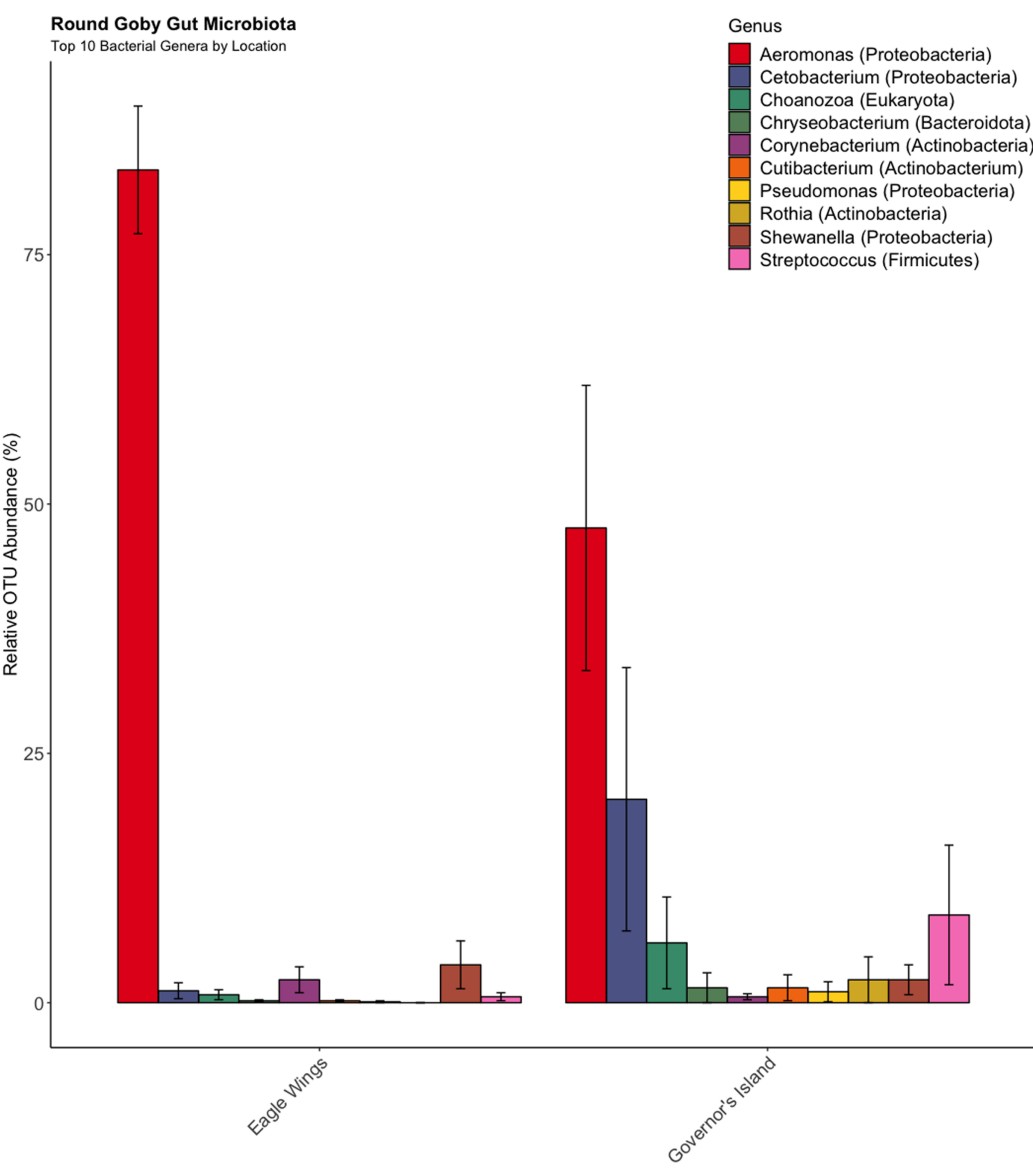

**Figure 5 Relative abundance of Top 10 OTU's for Round Goby by location from the habitat comparison.** *Aeromonas* spp. dominated the gut microbiota of Round Goby at the Eagle Wings, while the gut microbiota of Governors Island Goby was spread out across multiple genera including *Aeromonas* spp., *Cetobacterium* spp. and *Streptococcus* spp.

bacterial community between species (RG: $t$ = -4.11, df = 293.37, $P$ < 0.001 and YB: $t$ = 3.20, df = 84.47, $P$ = 0.002) compared to within species (Fig. 8). For Round Goby, additional Unifrac distance testing indicated a significant difference in bacterial community variation between habitats ($t$ = 11.30, df = 50.49, $P$ < 0.001). Round Goby from the deepwater habitat (Eagle Wings Islands) demonstrated the least amount of variation in gut microbial composition while those from the nearshore littoral Governors Island had the greatest amount of variation (Fig. 9).

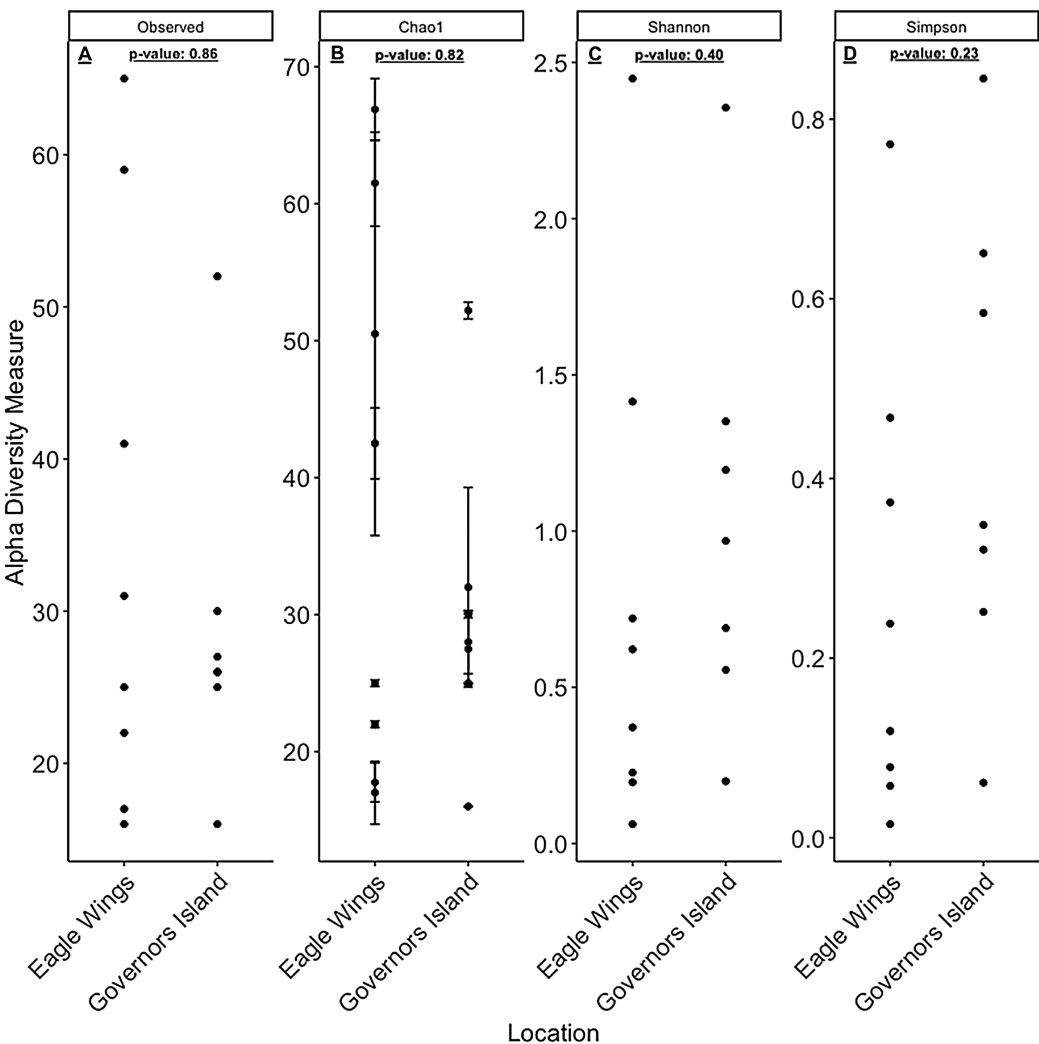

**Figure 6 α-diversity metrics calculated from the Round Goby vs. Yellow Bullhead species comparison.** (A) Observed, (B) Chao1, (C) Shannon and (D) Simpson α-diversity indices showed no differences between species when tested via Wilcoxon rank-sum tests (Mann–Whitney) (all *P*-values > 0.18).

## Comparing round goby and yellow bullhead gut microbiota predicted functions using Tax4Fun

Predictions of microbial community functions using KEGG Orthologs (KO) indicated no significant differences in predicted microbial functions associated with our species and habitat comparisons. Mapping of the Top 20 KO's that matched to KEGG pathways revealed only slight differences in microbial community gene abundances linked with environmental information processing, cellular processes, metabolic pathways, and genetic information processing (Figs. 10 and 11). There were no significant differences in the top 20 KO's for the species comparison for environmental information processing ($t = 0.28$, df = 9.22, $P = 0.79$), cellular processes ($t = 0.18$, df = 10.07, $P = 0.86$), metabolic pathways ($t = 0.58$, df = 19.75, $P = 0.57$), and genetic information processing ($t = -0.19$,

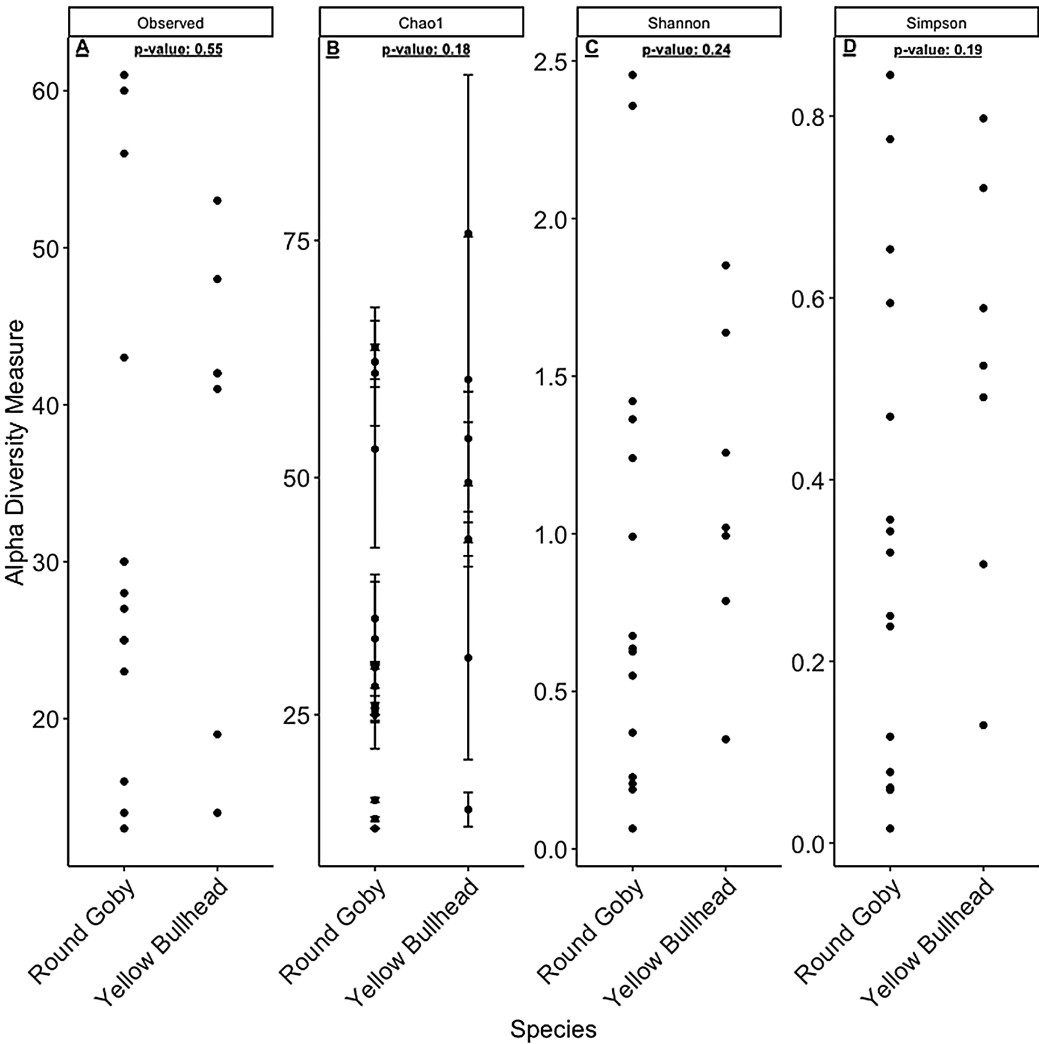

**Figure 7** α-**diversity metrics calculated from the Round Goby habitat comparison.** (A) Observed, (B) Chao1, (C) Shannon and (D) Simpson α-diversity indices showed no differences between habitats when tested via Wilcoxon rank-sum tests (Mann–Whitney) (all *P*-values > 0.46 ).

df = 9.36, *P* = 0.86). Similar conclusions were drawn from analysis of the top 20 KO's for the habitat comparison for Round Goby (environmental information processing: *t* = −1.86, df = 7.00, *P* = 0.11; cellular processes: *t* = 0.18, df = 10.07, *P* = 0.86; metabolic pathways: *t* = 1.93, df = 7.13, *P* = 0.09; genetic information processing: *t* = 1.75, df = 7.05, *P* = 0.12). This all stands in contrast to the aforementioned significant differences in overall gut microbiota both between and within species as seen through our β-diversity analyses.

## DISCUSSION

Our data revealed fish gut microbiota differences between invasive Round Goby and native Yellow Bullhead in the upper St. Lawrence River. Our findings add to previous research that describes microbial community dissimilarities in various fish species employing a

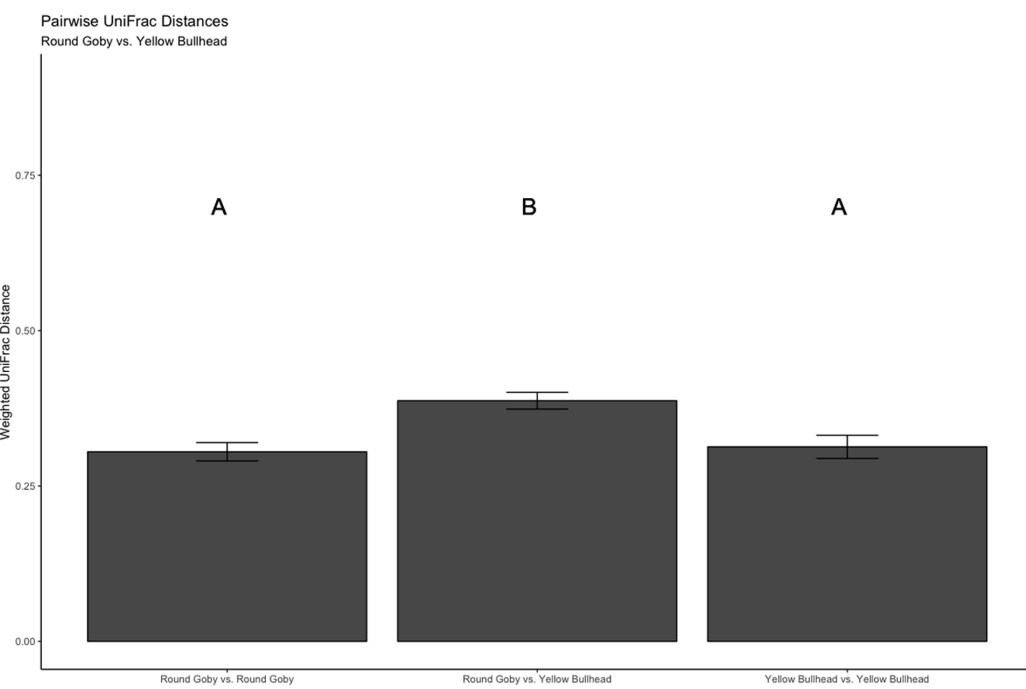

**Figure 8 Pairwise UniFrac distances calculated from the OTU's derived from the species comparison.** The UniFrac analysis revealed higher variation in gut microbial communities between species than within species. Differences between letters above each barplot denote a significant difference in UniFrac distance (*P*-value < 0.05).

range of sequencing methods (Denaturing Grade Gel Electrophoresis (*Li et al., 2012*); Pyrosequencing (*Li et al., 2014*)) as well as amplifying different variable regions of the 16S rRNA gene (V4 region; *Liu et al., 2016*). Both NMDS and PCoA analyses support overall microbial composition differences between these fish species and results from the SILVA database indicated dominance of specific bacterial genera in each species' gut microbiome. Additional hypothesis testing using PERMANOVAs and weighted Unifrac distances further verified that significant differences in β-diversity were present both between Round Goby and Yellow Bullhead, and within Round Goby caught at the Eagle Wings and Governors Island.

The observed prominence of *Aeromonas* spp., *Cetobacterium* spp., and *Clostridium* spp. in the fish gut is consistent with reports by others including *Aeromonas* spp. (*Wu et al., 2013*; *Li et al., 2016*), *Cetobacterium* spp. (*Li et al., 2015*) and *Aeromonas* spp. and *Cetobacterium somerae* (*Larsen, Mohammed & Arias, 2014*). In addition to microbial differences noted between fish species, we also report interspecies variation in gut microbiota between the same species (Round Goby) captured in different habitats separated by a water depth gradient. Simulation analysis from *Anderson & Walsh (2013)* suggests PERMANOVA analysis is robust even with heterogeneity of variance between groups under a balanced design. Larger scale sampling from multiple locations with a balanced sampling design is recommended to better understand patterns of within-species differences in microbiome among habitats. Future studies should take into account size and/or age bias with Round Goby, given the differences in average size of the captured

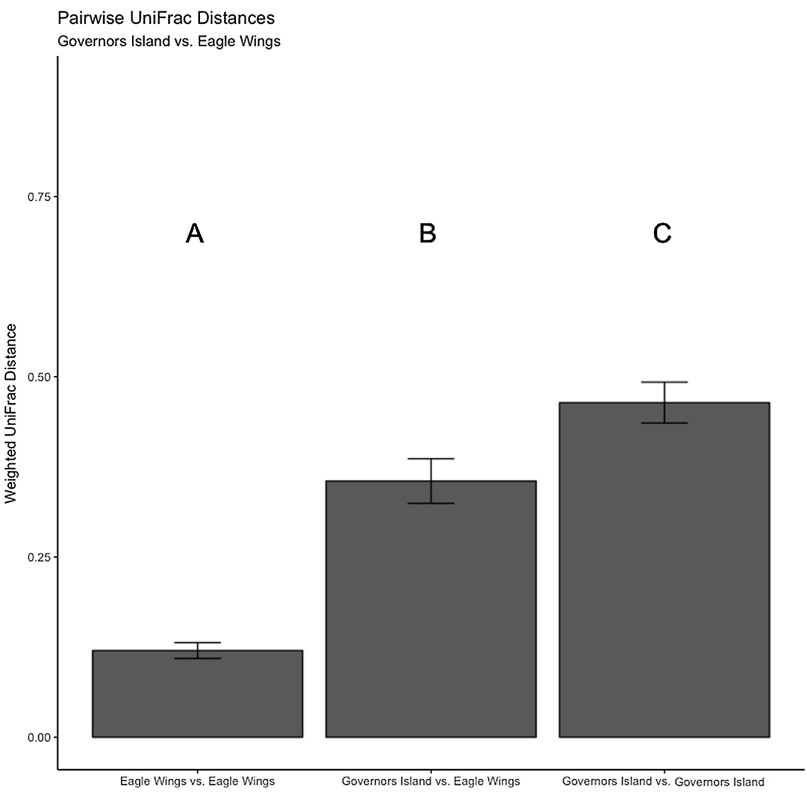

**Figure 9 Pairwise UniFrac distances calculated from the OTU's derived from the Round Goby habitat comparison.** Round Goby gut communities differed between habitats and deepwater (Eagle Wings Island) fish had less within habitat variations than fish in the near shore littoral (Governors Island) Differences between letters above each barplot denote a significant difference in UniFrac distance (*P*-value < 0.05).

Round Goby by habitat (150 mm at Governors Island vs. 113 mm at the Eagle Wings). Previous research on Zebrafish has indicated that gut microbiota can vary with age/development (*Stephens et al., 2016*) and by seasonal sampling (*Naviner et al., 2006*). All fish associated with this study were captured within an 11-day period during the fall of 2017, however observed differences in gut microbiota may not be present throughout all seasons. These variables will need to be controlled in future studies that increase sampling frequency spatiotemporally to extend our finding in a simple case.

The presence of specific and novel OTUs is particularly interesting when hypothesizing the microbiota's role in the gut community and its effect on fish hosts. For example, some species of *Aeromonas* act as opportunistic pathogens and cause various hemorrhagic fish diseases like furunculosis (*Hidalgo & Figueras, 2012*). Although *Aeromonas* spp. were routinely detected in fish guts in this study, we observed no obvious tissue hemorrhaging. However, we were unable to resolve *Aeromonas* sequences to the species level and were not able to determine if the specific *Aeromonas* bacteria present were known fish pathogens. This was most likely due to high heterogeneity of the 16S gene in *Aeromonas* bacteria (*Janda & Abbott, 2007*). Sequencing with additional universal gene

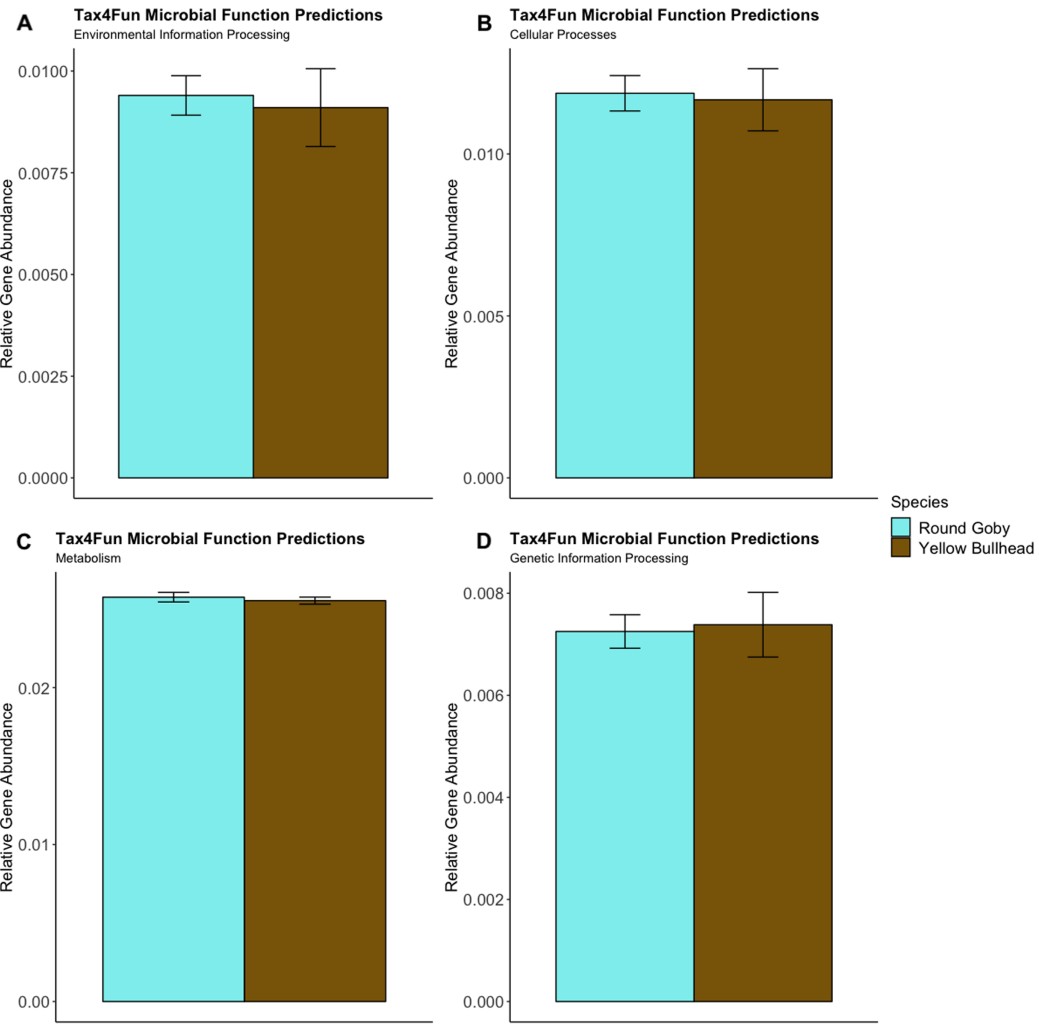

**Figure 10 Relative Gene Abundance of predicted microbial community functions from the species comparison, calculated using Tax4Fun and the Top 20 level one pathway KEGG Orthologs.** There were no significant species differences in microbial community functions pertaining to Environmental Information Processing (A), Cellular Processes (B), Metabolism (C) or Genetic Information Processing (D) for all measured level one KEGG Ortholog pathways (*P*-values = 0.79, 0.86 and 0.57, respectively).

primers (e.g., *cpn60* (*Minana-Galbis et al., 2009*)) may help reveal species delineation in *Aeromonas* bacteria. Additional microbial genera detected in this study are also known to be beneficial and/or detrimental to their hosts. For example, bacteria of the genera *Clostridium* produce various essential fatty acids and vitamins (*Ringø, Strøm & Tabachek, 1995*; *Givens, 2014*) despite some species like *Clostridium difficile*, which can opportunistically cause pseudomembranous colitis (*Kelly, Pothoulakis & LaMont, 1994*). Moreover, the bacterium *Cetobacterium somerae*, the most abundant genera discovered in the gut microbiome of Yellow Bullhead in our study, is known to synthesize vitamin B12 in the fish gut (*Tsuchiya, Sakata & Sugita, 2007*). Regardless, 16S predictive functions derived from our Tax4Fun analyses were unable to identify microbiota functional patterns between Round Goby and Yellow Bullhead and within Round Goby by sampled habitat.

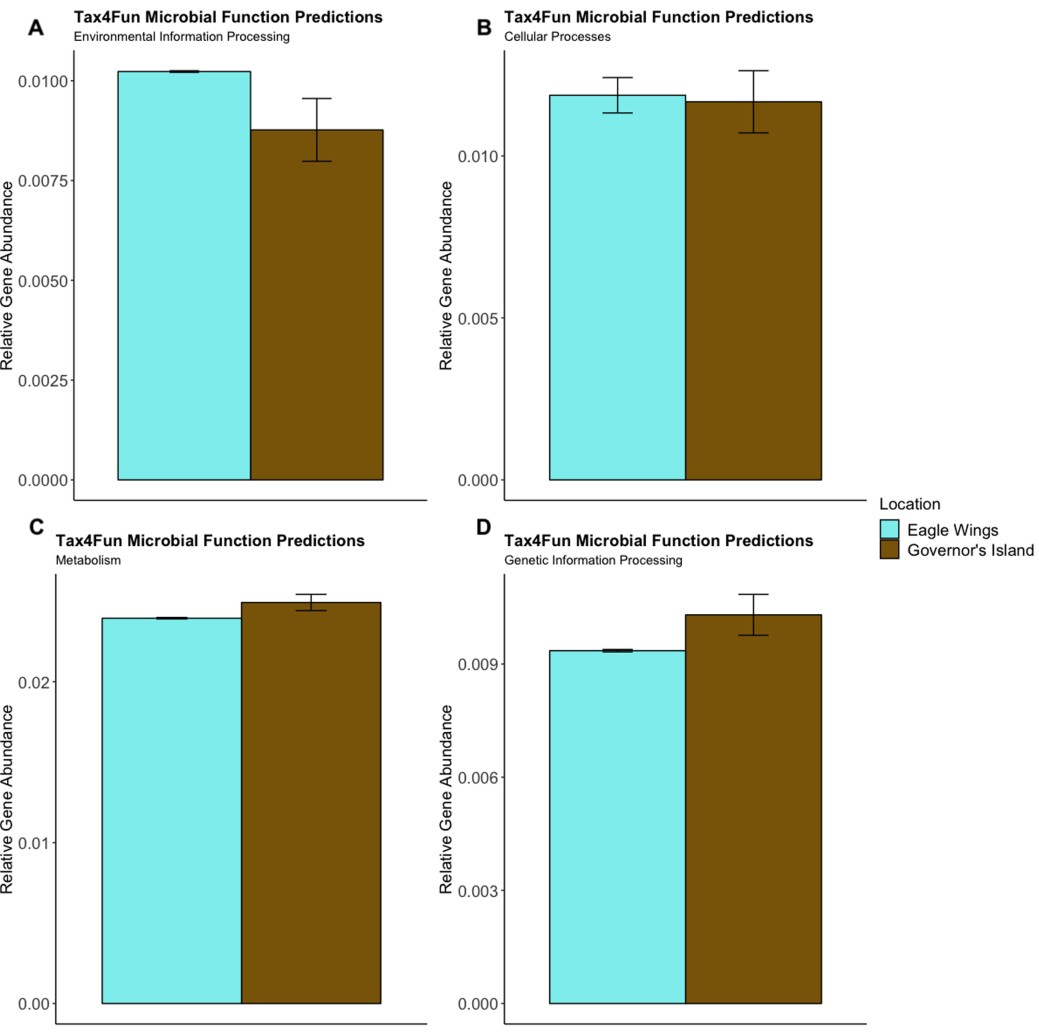

**Figure 11 Relative Gene Abundance of predicted microbial community functions from the habitat comparison, calculated using Tax4Fun and the Top 20 level one pathway KEGG Orthologs.** There were no significant habitat differences in microbial community functions pertaining to Environmental Information Processing (A), Cellular Processes (B), Metabolism (C) or Genetic Information Processing (D) for all measured level one KEGG Ortholog pathways ($P$-values = 0.11, 0.09 and 0.12, respectively).

In order to gain a better understanding of potential gut microbiota function in the fish gut, sampling with shotgun sequence metagenomic analyses could be implemented with parallel 16S amplicon sequencing analysis. The 16S amplicon sequencing would help delineate relative abundance of microbiota, and metagenomic analysis would define individual microbes' complete genomes. Together these approaches may provide a more complete picture regarding the metabolic, pathogenic, and other characteristics associated with bacteria sampled from the fish gut. The need for metagenomic analysis in has been called for in recent reviews on the fish gut microbiome (*Tarnecki et al., 2017*; *Gallo, Farrell & Leydet, 2020*) and implementation will almost certainly provide a better understanding on gut microbial functions that may significantly influence fish health.

In addition to the demonstration of microbial differences across species, NGS has the potential to detect minute differences in microbial communities within species. We employed an NGS workflow to investigate differences in gut microbiota in Round Goby samples from different habitats separated by less than 750 m. Evidence for environmental differences in fish gut microbiota has been previously noted in the Atlantic salmon parr gut microbiome (*Dehler, Secombes & Martin, 2017*) and in wild vs. aquaculture-reared Fish Species (*Ramírez & Romero, 2017*). Although our design did not take into account potential diet and habitat variation between the two samples of Round Goby, our data does support within-species differences at fine-spatial scales. A previous study in a nearby area of the St. Lawrence River indicated significant differences in total phosphorus and crustacean zooplankton abundance between shallow and deepwater stations (*Farrell et al., 2010*). These differences possibly extend to benthic invertebrates, including dressenid mussels known as important forage for Round Goby. Both Round Goby habitats were sampled at different depths (Governors Island: <2 m; Eagle Wings Islands: >15 m) and significant gut microbial community differences were noted, specifically with *Aeromonas* spp. (>80% total Eagle Wings Islands reads vs. ~50% of Governors Island reads). Nonetheless, α-diversity metrics indicated no differences in gut microbiota abundance in both our habitat and species comparisons. These analyses support that despite overall differences in the taxonomy of the bacteria, the diversity and evenness of the microbial communities within compared groups remained similar. The similarity in α-diversity metrics especially within the species comparison may point to limited ecological niches for microbiota to inhabit in the fish gut. Therefore even though the microbiota identity differs between species, Bacteria having similar metabolic capacities/functions in the gut environment prevent large fluctuations in α-diversity between fish species. Future analysis derived from NGS microbial data coupled with biological (e.g., host diet, host habitat) and environmental factors (e.g., pressure, temperature and light penetration) would help further define gut community structure differences observed between habitats and fish species.

## CONCLUSIONS

In this study, we employed an NGS workflow and sampled gut microbiota for the first time from fishes along the upper St. Lawrence River. We implement analyses to test both inter-and intraspecific differences. We describe significant species level microbial community differences between an invasive (Round Goby) and native (Yellow Bullhead) fish species. We also describe significant differences between Round Goby sampled in different habits separated by only ~750 m. This detailed workflow may help facilitate greater application of gut microbiome research and allow for the examination of numerous basic and applied questions (*Gallo, Farrell & Leydet, 2020*). As NGS technologies and knowledge of host-microbe interactions continue to grow, investigations into the gut microbiome will undoubtedly improve our understanding of fish ecology and their conservation and management. The research methods developed and applied here are intended to promote such investigations, and aid researchers interested in studying gut microbiota.

## ACKNOWLEDGEMENTS

A special thanks is given to the managers, postdocs, fellow graduate students, and technicians at Thousand Islands Biological Station that aided in collection of fish samples. We would also like to thank Dr. Donald Walker from Middle Tennessee State University and Dr. Karine Leydet from Syracuse University for providing a review prior to submission. This research is a contribution of the Thousand Islands Biological Station.

### Funding

Funding support was provided by the Great Lakes Research Consortium (GLRC) grant #17072612, the New York State Environmental Protection Fund grant AMO#10165 administered by the NYS Department of Environmental Conservation, and the SUNY Center for Applied Microbiology. The funders had no role in study design, data collection and analysis, decision to publish, or preparation of the manuscript.

### Grant Disclosures

The following grant information was disclosed by the authors:
Great Lakes Research Consortium (GLRC): #17072612.
New York State Environmental Protection Fund: AMO#10165.
NYS Department of Environmental Conservation.
SUNY Center for Applied Microbiology.

### Competing Interests

The authors declare that they have no competing interests.

### Author Contributions

- Benjamin D. Gallo conceived and designed the experiments, performed the experiments, analyzed the data, prepared figures and/or tables, authored or reviewed drafts of the paper, and approved the final draft.
- John M. Farrell conceived and designed the experiments, analyzed the data, prepared figures and/or tables, authored or reviewed drafts of the paper, and approved the final draft.
- Brian Leydet conceived and designed the experiments, analyzed the data, prepared figures and/or tables, authored or reviewed drafts of the paper, and approved the final draft.

### Animal Ethics

The following information was supplied relating to ethical approvals (i.e., approving body and any reference numbers):

State University of New York College of Environmental Science and Forestry's Institutional Animal Care and Use Committee approved the study (180202).

## Field Study Permissions

The following information was supplied relating to field study approvals (i.e., approving body and any reference numbers):

All specimens were collected through volunteer angling, baited minnow traps, or fine mesh hoopnets and under permit from the New York State Department of Environmental Conservation (license #354).

## Data Availability

Detailed code and protocols are available at GitHub: https://github.com/bngallo1994/Benjamin-Gallo-16S-Gut-Microbiota-Standard-Operating-Procedures-and-Associated-R-code.

Raw sequence reads are available at NCBI GenBank: PRJNA528762.

## Supplemental Information

Supplemental information for this article can be found online at http://dx.doi.org/10.7717/peerj.10237#supplemental-information.

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
