# Peer review of "Use of next generation sequencing to compare simple habitat and species level differences in the gut microbiota of an invasive and native freshwater fish species"

_PeerJ, doi:10.7717/peerj.10237_

## Round 0.1 · original submission · Major Revisions

· Academic Editor

Major Revisions

The reviewers generally agree that the work is of value, and while not necessarily broad, novel or rigorous to make a significant advance in the field, the results are properly communicated and appropriate for publication in PeerJ. All of the reviewers agree, and I concur, that the manuscript seems to be bifurcated in emphasizing the microbiome results relative to the novel DNA extraction and preservation methods. The former is less novel, but correctly presented, while the latter is exciting and novel but has not been thoroughly tested in the manner necessary to make it broadly adoptable by the community. My strong recommendation is that you do everything you can to present results demonstrating rigorous evaluation of the methods and then complement this in the manuscript by making it a clear and significant piece, and then restructure the manuscript to clarify how the two aspects are parallel.

·

Basic reporting

The overall manuscript is written ok. The structure could be improved with a more in depth literature review as noted. Some incorrect uses of references. Please check references and perhaps add some additional ones focused specifically on fish microbiomes. For the last paragraph of the introduction, I would encourage having a couple of overview sentences describing the general sampling strategy to test your hypotheses. The figures and tables are ok. The author cites the github page in the abstract but the link shows that nothing is listed. This needs to be made publically available or at least available to reviewers during the review stage.

Experimental design

Primary research aim:
The primary research question was to what extent a new storage buffer enabled the preservation of the gut microbiome of the fish. The author claims the buffer can be used to store samples for 4-8 weeks without cold storage but does not actually test this. The main point of the article is that they've developed a method to enable collection and storage of gut microbiomes from the field.

Scope cont.: It would make for a more compelling manuscript to describe the extent of fish diversity which would benefit from this. In other words, of the 35000 fish species, how many of them are located in remote locations where sampling is perhaps even impossible. I realize it would be difficult to give absolute numbers here but it would be useful to consider describing this a little more in the introduction.

The authors decided to focus on the validation of the technique, but did not test this thus the investigation was not rigorous. The authors may perform this analysis and submit findings in a revision.

Validity of the findings

For microbiome analyses, the author omits alpha diversity from the comparisons between species and comparisons in habitat for the gobies. This should be performed and compared using a non-parametric pairwise comparison e.g. Mann-Whitney. This would likely reveal the bullhead fish having higher richness and within gobies likely no differences in richness when comparing the sites. For both figure 2 and figure 3, the author compares the species and habitats using NMDS. While this is a fine measure, I would suggest and encourage the authors to explore additional beta diversity comparison methods which take into account phylogenetic differences in the microbial community (e.g. Unifrac distances). At the very least, utilizing methods such as Jaccard and or Bray-Curtis would enable additional comparisons which might further support the observations made. Figures 4 and 5 could be combined into one figure with two panels. Its not appropriate to just pick the top 3 OTUs. Also note that compositionality makes it impossible to determine true biomass or even changes or differences in communities. For instance, its possible that Aeromonas is higher in gut samples from Eagle Wings compared to Governors Island, but its equally likely that there are simply more other types of taxa in Governor’s Island and Aeromonas is the same. To deal with this problem, one can use ranks or ratios.
See, https://doi.org/10.1016/j.annepidem.2016.03.003

Additional comments

Developing new collection methods is important and while you have shown that you can distinguish a species effect and habitat effect based on samples collected with your method, you do not actually verify or test the method itself. If the focus of the manuscript is to be on the method, you will have to perform experiments which justify the buffer use. This could include a time series of a few samples processed with different storage methods. You can then compare both alpha and beta diversity changes. Alternatively, the dataset in itself has relevant biological findings which could also be a main focus rather than the storage method. That being said, it would still be important to justify this storage method and confirm that the differences in community structure are not simply from a storage effect.

Reviewer 2 ·

Basic reporting

Gallo et al investigated the gut microbiota of a native an invasive fish species collected from the St. Lawrence River using a 16S-amplicon sequencing approach. As the author emphasized in the introduction, investigation of the gut microbiota of fish is emerging in the filed of the Gut microbiome and it is necessary to understand microbial ecology of fish in the filed of the vertebrate gut microbiome. Although the data presented provided in this manuscript is quite simple, however, all the procedure from sampling to bioinformatic analysis follows the standard methodologies which I see as a strong point of the manuscript.

Only one point that needs to be clarified in the introduction and discussion is what the authors suggested from this study. If this manuscript aims to provide an overview of the gut microbiome composition and structure in two selected fish species (an exploratory study) the data that have been presented are relevant. However, from some part of the manuscript (i.e. line 79-82), it seems that authors suggesting the methodology used in this study as a standard SOP for all fish related gut microbiome analysis, which I believe is an oversimplification. Just as an example, although NAP buffer showed some promising results in one study for mammalian samples (Camacho 2013), it does not mean it provides the same good preservation picture for fish gut digesta. So, I believe unless you a clear comparison of the current state of art approach like snap frozen, RNALater, Omnigen buffer, Zymo storage buffer, etc, the conclusion that the author made in this manuscript is not robust enough. The same for DNA extraction.

Experimental design

As I explained partly in the Basic Reporting section, If we consider this manuscript as a preliminary exploratory analysis effort (and note a methodological suggestion paper), the experiments were done in a proper way. However, some improvements can be done.
1. One important point to mention for this section is the low number of fish selected for this analysis. I believe this is one of the key limitations of the manuscript that should be mentioned in the discussion section( e.g why this sample size).
2. Another type of analysis that could have added more information and insight to this manuscript is to use the functional prediction tool on their 16S dataset to have also an overview of the functional profile of the fish gut microbial communities. I suggest that given to reliable bioinformatics tools that are available for this purpose, the authors follow up on that and add the relevant multivariate analysis and discussion, too.

Validity of the findings

Given the high interindividual variation in animals such as fish, choosing a proper sample size is a critical factor for the microbiome analysis. While the methodology used in this manuscript followed the state of art approaches, the small sample size selected in this study could be a problem for reproducing the same findings by another research.

Additional comments

In General, I believe this study was a nice effort to provide a preliminary overview of the gut microbiome in two native and invasive freshwater fish species in St. Lawrence River and to show how in general the gut microbial community affected by the different habitats. Although the study suffers from a small sample size, however, the overall methodology was according to current standards in the field of microbiome research. By adding the prediction of functional annotation and relevant analysis and discussion to the current manuscript, I believe the manuscript can be accepted for publication.

Reviewer 3 ·

Basic reporting

The manuscript is a well-written and useful summary of the current status of “data publication” from a certain perspective. The authors, however, need to be more analytical. This manuscript does an excellent job showing a new protocol that can be easily used to take fish gut samples in the field that can be used to gut microbiome analysis without using -80 oC to preserve the samples until further microbiome analysis. The paper would be both more compelling and useful to a broader audience if the authors provide a better picture of the factors that drove the gut microbial community in fish as well as the importance of the host’s physiology and health.

Related to the title, it is suggested to modify the title. NGS has been used widely in gut microbiota studies in fish, the main focused on the paper is the NAP buffer procedure used to eliminate the need for refrigeration of samples at -80oC. The title should be more tied to that.

In the introduction, there are some facts related to the factors that address the microbial community in the fish gut, as well as, the effect in the host physiology and health. A suggestion, all the paragraphs should be reorganized as there are some sentences that are very redundant. The statement of the factors leading the gut microbiome composition and colonizing needs more facts to support it.

Experimental design

The section material and method are very clear, however, along with the description there is some lack of information that requires description:
It is important to describe the sampling time and the season. In the case of environmental studies of the gut microbiome in fish, the season time has shown to have a very important effect on the feeding behavior of the animals. As well, if the animals were sampled at different days or different sampling times, they could have been affected by the digestion process as the animals in nature compared with cultured fish have their feed all time.

As well, to know the size of the animals caught would be very helpful in order to have an idea of the life stage of the animals studied which complements the fact of their different trophic feeding behavior.

The paragraph between line 189 to 191, the statement of the Atlantic salmon: This statement requires to be rephrased. The information is a little confusing. Inline 54, eliminating the word can. In the sampling procedure, did you consider sampling some water?

Validity of the findings

The discussion of data citation was good and captured the state of the art well, but again I would have liked to see more discussion on their findings.

In the discussion, the authors might consider some questions: 1)if the 2 euthanasia protocols used in the sampling time could affect the gut microbiome composition. It has been shown that the psychical stress caused by human handling can affect the gut microbiome in fish.
2) The possible effect due to washing and removal of the digestive feed on the DNA extraction as well need to be considered. It has been shown that this could cause some troubles in the bacterial DNA extraction in gut microbiome studies.

How could it be explained why the microbial composition in the governor’s islands is so disperse compared to the eagle wings? As well, why is it so abundant the presence of bacterial strains like Aeromonas and Clostridium that has been associated with fish disease?

The possible differences between Round Goby at different depths should consider the effect of the two different sample points besides the depth. It will be very interesting to study if the NAP buffer will give the same results in the same samples at 4 weeks or 8 weeks prior to DNA extraction.

Prior to reading the manuscript, I thought the results section of the paper was going to be promising and relevant, but it is somehow light information. It would be better if the richness and diversity identified were presented on a table with other diversity indexes like Shannon, Simpson, etc. As well, a description of which bacteria genus shared between fish species and within species should be presented. How many OTUs they share as well. Figures 2 and 3: a) rarefaction curves require the legend reflecting sample label.

In the supplemental figure S2, the final word to describe will be location or habitat?. Finally, PCO plot graph seems better to describe the differences between species and habitat instead of a NMDS. The R2 and stress value is not the strongest in both compared factors.

Additional comments

The manuscript is a well-written and useful summary of the current status of “data publication” from a certain perspective. The authors, however, need to be more analytical. This manuscript does an excellent job showing a new protocol that can be easily used to take fish gut samples in the field that can be used to gut microbiome analysis without using -80 oC to preserve the samples until further microbiome analysis. The paper would be both more compelling and useful to a broader audience if the authors provide a better picture of the factors that drove the gut microbial community in fish as well as the importance of the host’s physiology and health.

---

## Round 0.2 · Minor Revisions

· Academic Editor

Minor Revisions

Thank you for your patience with this manuscript...we had some challenges with obtaining proper reviews on the revisions. I am pleased that I was able to review your manuscript thoroughly and consider it acceptable upon minor revision following the suggestions of the reviewer.

Reviewer 3 ·

Basic reporting

The overall manuscript is good. The structure could be improved with a more in-depth literature review as noted. Some incorrect uses of references and is important homogenizes the terms like gut microbiota or gut microbiome along the document. Please check references and perhaps add some additional ones focused specifically on fish microbiomes in wild based on trophic habit and feeding strategies (i.e. The gut microbiome and degradation enzyme activity of wild freshwater fishes influenced by their trophic levels by Liu et al 2016). The tables and figures are ok. The github page in the abstract cited by the author shows that nothing is listed. This needs to be available to reviewers during the review stage.

Experimental design

The primary research aim should be very clear: you proposed a gut microbiome analysis comparing two different species (feeding strategies and locations-which this part is missing), and only compare habitat within the same fish species (RG).

When they sampling the fish, they say the animals were euthanized within ~ 3 hours of capture; did they were kept together in the same container sharing the water during that time individually until euthanize and dissection?

Back in the preservation method, although NAP buffer showed some promising results in one study for mammalian samples (Camacho 2013), it does not mean it provides the same good preservation picture for fish gut digesta.

Line by line review
Line 173 correct the upperSt; line 318 punctuations; correct parenthesis in lines 443 and 727.
Line 732, never mention Cetobacterium somerae in the results (check this part)
For evenness should be calculated Shannon-evenness not only Shannon index line 781
Table 1 & 2 indicated in line 389 only shows one complete data set of coverage (description as well only say Table 1)

Validity of the findings

The calculation of alpha and beta diversity improves the results in the manuscript. There is some suggestion about how the alpha diversity index are presented. It would be better if the index results (figure 6 and 7) were presented on a table and diversity index like Simpson, and evenness index like Shannon-evenness could be added to it. How could you explain that the two different species didn’t show any significant difference among the 3 alpha diversity indexes?
Figure 3AB: What would be the main reason that samples of the eagle wings are less disperse than the governors island. It will be interesting determining the number of OTUs shares between habitat and species, as well which genera are the ones that are unique in each fish species.

Additional comments

With the changes made on the original manuscript clarify better the aim of it. The authors focused on pinpoint that there is significant difference between gut microbiome. Adding the PCO ordination, you can distinguish better a species effect and habit effect on GO based on samples collected with your method.

---

## Round 0.3 · accepted · Accept

· Academic Editor

Accept

Thank you for your revisions. We now consider the current draft acceptable for publication in PeerJ. All the best to you and thank you for considering the journal for your work.